# The Impact of the Implementation of Preventive Measures Due to COVID-19 on Work Design and Early Childhood Professionals’ Well-Being—A Qualitative Study

**DOI:** 10.3390/ijerph19031739

**Published:** 2022-02-03

**Authors:** Susan Gritzka, Peter Angerer, Reinhard Pietrowsky, Mathias Diebig

**Affiliations:** 1Institute of Occupational and Social Medicine (IASUM), Centre for Health and Society (CHS), Faculty of Medicine, Heinrich Heine University (HHU) Düsseldorf, Moorenstraße 5, 40225 Düsseldorf, Germany; peter.angerer@hhu.de (P.A.); mathias.diebig@hhu.de (M.D.); 2Institute of Experimental Psychology, Department of Clinical Psychology, Heinrich Heine University (HHU) Düsseldorf, Universitätsstraße 1, 40225 Düsseldorf, Germany; r.pietrowsky@hhu.de

**Keywords:** child care, occupational health and safety, COVID-19, preventive measures, well-being, work demands, early childhood professionals

## Abstract

The reopening of child-care programs during COVID-19 demanded comprehensive preventive measures. Research to date has overlooked this reopening process as well as early childhood professionals’ (ECPs) implementation efforts and resulting changes in their work practices and well-being. As a result, this study sought insights into (1) the practical implementation of measures, (2) perceptions and evaluations of measures, (3) changes in work characteristics, and (4) its impact on well-being. Qualitative interviews were conducted with German child-care managers (*N* = 27) between June and August 2020. The semi-structured interviews were audio-recorded, transcribed, and content-analyzed using MAXQDA. ECPs, through a combination of high effort and engagement, ensured the feasibility of most preventive measures. This included practices which were perceived to be unreasonable or ones which were stricter than practices required for the public. This exacerbated the critical work characteristics (e.g., high workload, overtime, and multitasking) from pre-pandemic scenarios and led to new work demands (e.g., changes in work content and social interactions). ECPs maintained intensive work demands and consequently suffered from broad strain outcomes (e.g., worry, exhaustion, anger, fear of infection, and reduced psychological sense of community). This study highlights the adverse psychosocial work environment of ECPs despite the necessity of ensuring health and safety at work.

## 1. Introduction

### 1.1. Background and Current Research

At the onset of the COVID-19 pandemic, all child-care providers in Germany were mandated to close in March 2020 as an immediate response to the crisis. Initially, only children of essential workers relying on child care were able to attend throughout an emergency period. Following this initial three-month lockdown, child-care centers resumed operation with access for all children under specific health and safety requirements and other restrictions. Hence, the majority of early childhood professionals (ECPs) returned to their workplace. In advance of reopening, child-care providers received protocols for operating child-care programs during COVID-19 from different sources (e.g., state/federal government, public health authorities, and direct employers) in order to safely recommence during the continued presence of COVID-19. Those child-care policies enumerated various occupational health and safety (OHS) measures as well as infection prevention and control (IPC) practices targeted at the reduction in SARS-CoV-2 infection and transmission. Appropriate preventive actions were endorsed for implementation in order to protect children, staff, and parents. Predefined measures and recommended strategies operated at the individual and environmental levels and included, but were not limited to: (1) introduction of engineering controls (e.g., fixed group settings by adopting the space design and physical barriers), (2) preventive behaviors and use of Personal Protective Equipment (PPE), and (3) changes in administrative and work practices (e.g., reduction in child care hours, drop-offs/pick-ups outdoors, stricter management of sickness absence, enhanced hygiene practices, regulations regarding children’s play/catering/sleep, etc.).

Undoubtedly, the adoption of preventive measures in the workplace are a key factor that facilitate reopening after lockdown, yet they provide a myriad of challenges [1,2]. Effective public preventive measures for COVID-19, such as physical distancing and face masks [3,4] may only be partially feasible within a child-care setting due to the importance of face-to-face interaction for infants [5]. Thus, professionals are conflicted as they are required not only to follow standard preventive measures when interacting with parents or colleagues, but they also must adapt their behaviors and activities to meet the care needs of children at the same time. This scenario may substantially transform the nature of work in the child-care sector, thereby creating new workplace hazards [6]. This is of considerable importance since the work characteristics in child care were previously noted in the international scientific literature (i.e., prior to COVID-19 pandemic) to be psychologically and physically demanding [7,8,9,10]. This might be further aggravated by the ongoing pandemic [11]. Specifically a number of quantitative studies identified preexisting shortcomings and work demands, such as excessive group sizes, staff shortage, multitasking, insufficient breaks, high physical demands, inadequate time for planning and preparation, ergo, time pressure [12,13,14,15,16,17,18,19]. Besides these stress-eliciting factors of the work environment, the workplace is generally associated with an increased exposure risk and risk of infection [20]. Thus, in a recent study ECPs reported high fear of SARS-CoV-2 infection and fear of SARS-CoV-2 transmission [21]. Preliminary qualitative research with child-care managers identified additional, role-specific work demands. Among others, these involved growing demands of parents and politics in the absence of professionals’ participation, role diffusion (manager vs. colleague), lack of a management team (i.e., only one manager), difficult management of sickness absence and the need to react quickly to changes in daily routines [22]. The synergy of high work-related demands, psychosocial work demands and the low reward received, in terms of underappreciation by society [23] and low incomes [10], make ECPs vulnerable to the experience of stress [24,25]. Hence, long-term stress-related outcomes such as burnout [26,27] and depressive symptoms [28] have been found among the child-care workforce.

Collectively, the aforementioned studies can be explained by numerous organizational stress models and theoretical frameworks. These approaches aim to explain the occurrence of impaired health and well-being as well as their relationship with work characteristics. It has been shown that the interpretation of work characteristics as work demands may influence their relation with health [29,30]. Hence, how ECPs cognitively appraise situational demands of preventive measures mediates the influence of work characteristics and health [31]. Work demands refer to given work characteristics (e.g., physical, psychological, social, and organizational aspects) faced or experienced by the professional that come with prolonged physical or mental effort and result in physiological or psychological costs [32]. Among work characteristic models, the Job Demand-Control-Support model (JDCS model) [33,34] serves as one of the most influential, and extends the Job Demand-Control Model [35] by social support as a substantial work characteristic. The JDCS model predicts that the synergy of high job demands and low job control (i.e., ability to control work activities) lead to strain whereas social support acts as a moderator. Accordingly, social support buffers either the negative impact or additively impairs individuals’ well-being [36,37]. This is important since ECPs perceived low levels of control in their work, but also described team and leadership support as pivotal resources [38].

Until now, research has failed to comprehensively investigate the child-care work environment during the pandemic, but focused instead on the stress levels of parents and caregivers in need of child care during COVID-19 [39,40,41,42]. Moreover, recent pandemic-related stress and mental health research was predominately concerned with health care workers resulting in ample literature [43,44,45,46,47,48,49,50,51,52]. While surveys endeavored to holistically explore ECPs’ work practices and well-being during the pandemic, they lacked scientific rigor [53,54]. To date, only one quantitative study attempted to capture the psychological impact of COVID-19 on ECPs; thereby, reporting increased workload, work-related stress and concerns for children and families [55]. However, the specific sample characteristics remain unclear, since only one-third of the participants were directly employed at a child-care center. Two studies found that one-third of child-care directors showed clinically relevant levels of depressive symptoms during COVID-19 [56,57], while one study additionally emphasized an increase in depressive symptoms relative to the same population prior to COVID-19 [57]. However, both studies were based on child-care directors from centers in the United States where the majority of centers (53% and 75%, respectively) experienced mandated closures. In contrast, this study focuses on child-care centers that reopened during the pandemic.

In conclusion, ECPs’ work demands and well-being under COVID-19 has been overlooked to date in the scientific literature. There is a need to bridge this knowledge gap, as society depends upon ECPs given their status as essential workers [58]. Therefore, ECPs’ health must be a societal priority. Although preventive measures protect the physical health and safety, there might be potential unintended consequences of these virus mitigation strategies. According to the JDCS model, we assumed these preventive measures to be perceived as intensified efforts and alter the work characteristics leading to higher work demands. Similarly, the political demand to implement predefined measures might further reduce the individual decision latitude and social restrictions may diminish social support. Hence, to the best of our knowledge, this study is the first that explores ECPs’ experiences after lockdown that includes associations between implemented preventive COVID-19 measures, and changes in work characteristics and strain. Due to the originality of this research aim, we conducted a qualitative interview study with child-care managers in an exploratory manner. A qualitative narrative can provide necessary insights to facilitate a deeper understanding of the complex nature of work in child care during COVID-19.

### 1.2. Study Aims and Research Questions

In light of recent COVID-19 developments and imposed implementation demands, we firstly anticipated augmentations in efforts at work. Thus, the implementation of preventive measures requires sustained physical and mental effort. We therefore sought to gain knowledge on the practical implementation of OHS and IPC measures, which were adopted to ensure the reopening of child-care centers after several months of lockdown scenarios. The detailed description of how measures were actually implemented is essential to comprehend the subsequent answers to research questions 2, 3, and 4.

1.How are predefined and recommended preventive measures practically implemented in child-care centers and what changes and/or challenges occur?

Secondly, from a theoretical perspective, we aimed to understand the underlying thought process of child-care managers (i.e., cognitive appraisal) in order to investigate whether implementation efforts were perceived as burdensome. From a practical perspective, we strived to capture professional opinions on preventive measures to inform decisionmakers for future scenarios.

2.How do child-care managers perceive and evaluate preventive measures and the changes associated?

In a third step, we explored the influence of preventive measures on the prior work environment. We aimed to translate the perceived effort and work demand into certain work characteristics.

3.How do the preventive actions taken affect the work characteristics and its interpretation?

Drawing on the first three research questions, we finally investigated the individual strain reactions perceived by ECPs.

4.How do the preventive measures and COVID-19 influence the well-being of child-care managers and their staff?

## 2. Materials and Methods

### 2.1. Qualitative Study Design

A qualitative design was chosen to pursue an exploratory approach [59]. This method is particularly useful in applied health research, when the field of interest is understudied [60,61]. In order to adopt a systematic research process, we were guided by literature on qualitative exploratory research designs [62], data analysis, [63,64] and qualitative reporting [65].

### 2.2. Sampling and Recruitment Strategy

The study was approved by the local ethics committee (study number 2020-1067). In total, 371 child-care centers in a city in North Rhine-Westphalia, Germany, were contacted via e-mail in cooperation with the city’s youth welfare office in June 2020. Here, the job “child-care manager” refers to an individual who is responsible for the operational management of a child-care center (i.e., children prior to elementary school entry per German definition), the leadership of ECPs, the pedagogical concepts taught and the cooperation in the team, with parents and stakeholders. Unfortunately, we have no knowledge of how many child-care managers or deputies received the recruitment material. E-mails from the youth welfare office are not directly forwarded to child-care managers, but to their direct employers. Thus, most child-care centers are operated by nonprofit agencies (e.g., churches, municipalities, public schools, welfare, and government agencies) or privately operated for profit by organizational and individual owners. As a second recruitment action, we conducted telephone acquisition by calling randomly selected phone numbers of child-care centers that are openly accessible in an online child-care registry. To finalize purposive sampling, recruitment was ‘snowballed’ by asking those participating if they knew others who would be interested. Following the initial contact, participants received a comprehensive invitation, explaining context underpinning the study and the interview. Prior to interviewing, participants signed a declaration of informed consent. To be included, participants had to be a manager or a deputy in a child-care center in the respective city. We stopped sampling until we reached data saturation, thus further interviews delivered redundant information and failed to yield any new emergent themes [66].

### 2.3. Semi-Structured Interviews

Before commencing the study, three of the authors (S.G., P.A., and M.D.) analyzed relevant topics on pandemic-related guidance and policies in child care. A semi-structured interview guideline was designed according to research questions. This interview guide developed as follows: First, we collected all questions of interest for the research aims. Second, we scrutinized and reduced the questions in terms of openness, suggestibility, and importance. Third, the remaining questions were sorted and, fourth, subsumed in topics [67]. Two authors (S.G. and M.D.) pre-tested the applicability of the first interview guideline draft with two directors from the youth welfare office who have vast experience in the child-care setting and previous experience in managing child-care centers. Following minor adjustments, the first author (S.G.), working in the field of occupational health psychology, conducted 27 interviews in German with child-care managers between June and August 2020. Since social restrictions were present, interviews were carried out by telephone. The interview was opened with an information phase (i.e., introduction, information, and confidentiality) and a warm-up phase (i.e., description of child-care center and work activity) [68]. During the main phase, interview questions asked child-care managers to reflect on a typical day in the child-care center following the reopening. Furthermore, participants were encouraged to describe the experience of the implementation of pandemic-related measures. On that basis and whenever appropriate, the interviewer openly explored which measures are demanding, succeed well and if relevant which challenges arise. The interviewer probed potential implications in greater depth in response to participant’s answers. A final question probed other themes potentially relevant to the interviewee. At the end, when the recording finished, participants answered brief socio-demographic questions to capture participant age, sex, highest educational level attained, number of years in the job position, number of years worked in the center, and the district of the center. The interviews lasted 36 min on average (min: 20 min; max: 50 min).

### 2.4. Analysis

The interviews were audio recorded (Olympus LS-P1) and professionally transcribed verbatim by an external transcription service following the transcription rules by Dresing and Pehl [69]. Each transcription was checked for accuracy and complete anonymity (S.G. and H.T., see acknowledgments). Transcripts averaged 18 pages (min: 9; max: 33). In total 491 pages of interview material were reviewed. The qualitative content analysis [70] of the data was led by S.G., and performed using the software MAXQDA 2020 (VERBI GmbH, Berlin, Germany) [71]. A first sample of 30% (*n* = 8) of all final transcripts was double coded separately by S.G. and H.T. (see acknowledgments). Prior to this, the two coders agreed to use an initial coding scheme based on the four research questions in reference to the topics addressed in the interview guide; therefore, the four broader dimensions derived from the four research questions (i.e., applying deductive coding). Additional main and sub-categories were added continually by inductive category development by each coder. We applied summarizing as well as structuring content analysis simultaneously [64]. After this first individual coding, both coders compared their coding process. As a product of discussion and agreement, we developed a coding scheme that contained definitions, prototypical examples, and coding rules for each category (S.G. and H.T.). This first draft was additionally discussed with M.D. until we reached consensus between all three researchers. The coding process was further supported by a coding guideline containing general coding rules (e.g., coding sentence by sentence, assigning multiple codes to one quotation). In the next step, we independently applied this first draft of coding scheme to all interview transcripts, including the first eight interviews, to ensure data trustworthiness in terms of inter-coder reliability (S.G. and H.T.) [72]. We iteratively discussed new emerging codes, discrepancies, or overlapping themes and amended the categories and its definitions. This resulted in a final coding framework and confirmed data saturation (S.G., M.D., H.T.). In a final step, all transcripts were coded again by the first author by applying the final coding scheme consisting of three levels: (1) dimensions (research questions), (2) main categories, and (3) sub-categories. We provide quotations for each category in order to support the findings presented (see Appendix A). These were translated from German to English by a certified translator (L.G., see acknowledgments). The completed checklist of consolidated criteria for reporting qualitative research (COREQ) can be found in the Appendix A [65]. The socio-demographic data were analyzed for descriptive purposes in terms of frequencies, means, and standard deviations using IBM SPSS Statistics for Windows, version 25.0 (IBM Corp., Armonk, N.Y., USA) [73].

### 2.5. Study Participants

A total of 27 child-care managers (mean age = 48 years ± 10.76; range = 30–63 years; 93% female) accounting from all districts in the city were interviewed. Their mean length of experience in this job position varied from 0.25 to 34 years (mean = 9.21 years ± 10.26). Likewise, the mean length of employment at the current child-care center differed greatly from 0.5 to 29 years (mean = 9.97 years ± 9.3). Concerning the highest educational level attained, the majority of participants (67%) accomplished a child-care apprenticeship (i.e., 3–5 years in Germany); other participants had a college education (22%) or a university degree (11%). All child-care managers provided additional information about the total number of children they have the capacity to care for (mean = 68; range = 22–107) (i.e., not displaying the current number of children attending child-care during COVID-19) and the total number of ECPs (mean = 15; range = 3–22).

## 3. Results

The entire content analysis revealed four dimensions including the main and sub-categories illustrated in Figure 1: (1) implementation of preventive measures in child care and associated changes, (2) perception and evaluation of preventive measures, (3) impact on work characteristics and work demands, and (4) effects on ECPs’ well-being. Quotations that exemplify main and sub-categories are provided in the Appendix A.

### 3.1. Research Question 1: How Are Predefined and Recommended Preventive Measures Practically Implemented in Child-Care Centers and What Changes and/or Challenges Occur?

Participants described the practical implementation and adherence of multiple OHS and IPC measures. The summary of measures does not claim to represent the completeness of all measures required in a child-care setting due to COVID-19, but those acknowledged to be most relevant in a real-world setting. Preventive measures were adopted at all centers; however, the approach and execution were heterogeneous due to the different structural conditions and resources (e.g., size and location of the center, physical layout, staff situation, quality of parental cooperation, etc.).

#### 3.1.1. Reduction in the Duration of Child Care per Child (i.e., Reduction in Hours)

The weekly duration of child care was reduced by 10 h per child. Prior to reopening, child-care managers conducted surveys to clarify individual parental needs. The majority of this restructuring was parent-driven, with some children attending a full day less, and others reducing the hours per day. Child-care managers prioritized the needs of working parents. The hourly reduction seemed to be feasible without any difficulties, except for essential workers, since they had full child-care coverage during the emergency period. Finally, the child-care managers introduced fixed schedules to ensure staggered drop-off/pick-up times.

#### 3.1.2. Introduction of Fixed Group Settings

Child-care managers established fixed group settings in terms of a fixed composition (children and staff) as well as permanently assigned and used premises. These group settings had no direct contact with each other.

##### Conditions for Group Clustering

Interviewees reflected on numerous conditions that needed be considered for fixed group settings: (1) number of regular groups in the center, (2) age structures, (3) current number of children attending child care, (4) staff’s availability, (5) duration of child care, (6) parents’ preferences, (7) upcoming elementary school children, (8) siblings, (9) common travel routes (e.g., bus), and (10) emergency period groups.

##### Separation of Group Settings Indoors

Most interviewees were able to provide each fixed group setting with its own room and adjoining bathroom. If no separate bathrooms were available and/or accessible, bathrooms were used on a time-delayed basis, or use was agreed by telephone at short notice. Child-care managers allocated other rooms (e.g., gymnasiums and craft rooms) to group settings on an hourly or daily basis. ECPs mainly labelled common areas (e.g., corridor and cloakroom) with tape and colors for different group settings or physical barriers were installed.

##### Separation of Group Settings Outdoors

Child-care managers ensured the spatial separation outdoors by assigning individual plots. A few were able to set up access points for each group setting to the respective plot. Similar to indoors, markings and colors visualized the division in a child-friendly and playful way. Children with certain disabilities and special needs (e.g., visual or hearing impairments) had trouble in adhering to the spatial separation. Overall, activities were increasingly held outside. Surrounding nature and other outdoor playgrounds were exploited. Yet, a number of child-care managers criticized the small size of outdoor areas; hence, they time-staggered the usage. Others criticized outdoor play spaces to be function-specific (i.e., different areas for different activities) which did not allow spatial separation.

#### 3.1.3. Compliance with a Fixed Minimum Staffing Standard

Staff could no longer be exchanged between group settings and therefore flexibility was lacking and, conversely, more staff was necessary. Participants elucidated reasons why the minimum staffing standard derived from pre-pandemic child-care policies could no longer be fulfilled: (1) scarce staffing level prior to COVID-19, (2) illnesses, (3) greater sensitivity in dealing with symptoms of illness (i.e., preventive stay-at-home orders), (4) quarantine and/or long waiting for COVID-19 test results, and (5) high risk groups staying at home (e.g., pre-existing medical conditions and/or over 60 years old). Participants indicated factors that buffered staff shortages: (1) increased operational commitment of the child-care managers themselves, (2) strategic workforce planning, (3) fewer children in the center, (4) reduction in child-care hours, and (5) high staffing level prior to COVID-19.

#### 3.1.4. Implementation of Drop-Offs and Pick-Ups Outdoors

Child-care managers commonly focused much attention on drop-offs and pick-ups to avoid direct contact. Parents, guardians, and caregivers were restricted to enter the building, thus, interviewees organized drop-offs and pick-ups outside.

##### Spatial Distance during Drop-Offs and Pick-Ups

Child-care managers came up with innovative solutions to maintain physical distance: (1) one-way systems with signs and pictures (respecting illiterate adults or non-German speakers), (2) adults and/or children waiting areas labeled with the corresponding color of the group setting, (3) floor mats (1.5 m) as a visual demarcation, and (4) multiple entrances for different group settings (e.g., main entrance, emergency entrances/exits, gym doors). Due to local conditions, some child-care managers were not able to provide sufficient spatial distance and in addition introduced a staggered schedule.

##### Staggered Schedule during Drop-Offs and Pick-Ups

Taking the reduction in hours per child into account, half of the child-care managers staggered the children’s arrivals and departures either between all children, or even between group settings. Child-care managers extended the drop-off times beyond the original times and assigned specific times to each family upon prior agreement. However, in order to accommodate parents, child-care managers granted some time flexibility. The other half of child-care managers considered staggered schedules to be unfeasible or unnecessary. Reasons were: (1) sufficiently large outdoor areas to maintain physical distance, (2) common travel routes and arrivals (e.g., bus), (3) siblings in different group settings (i.e., parents did not drop off children twice), and (4) parents’ incapacity to stick to scheduled times due to various reasons (e.g., low engagement of families/parents).

##### Behavior of Staff and Parents in Drop-Off and Pick-Up Situation

Child-care managers reported that one designated staff member for each group setting was assigned to go outside, greet parents, pick-up the child and walk each child (respective to the children’s groups) inside the center, immediately followed by hand hygiene practices. During drop-offs and pick-ups staff wore masks and checked parents’ adherence to preventive measures. ECPs had to remind a few parents frequently of adequate PPE and preventive behaviors. Participants indicated the intentional reduction in communication between parents and staff as parents were obligated to leave the outdoor area quickly. Only a few exceptions allowed parents to enter the facility (e.g., settling-in period). Child-care managers encouraged parents to accompany their children with one consistent person nominated.

#### 3.1.5. Management of Sickness Absence and Testing for COVID-19

All child-care managers agreed on the importance of awareness and management of illness symptoms. They were aware of bearing the responsibility for the collectivity of children, families, and ECPs. Simultaneously, they emphasized the great difficulty in the practical handling of illness symptoms in the midst of a pandemic.

##### Symptomatic Children

At the time of reopening, child-care managers generally refused children to attend the child care as soon as they or people from the same household showed symptoms of any infectious illness or symptoms of COVID-19. ECPs immediately separated children who developed symptoms and informed parents. Participants communicated the dynamic nature of rapid changes in the official regulations as new available information on COVID-19 continued to evolve, which made decision-making challenging and led to uncertainty. Child-care managers explained that recommendations varied depending on the length of time a symptomatic child had to stay home (24 vs. 48 h symptom-free), type of symptoms (“colds” as COVID-19 symptom vs. not), and written certification of symptom-free children (physician vs. parent). In connection with these rapid regulatory changes, a variety of practical problems arose: (1) definition of a “sick child” (i.e., common for young children to have multiple respiratory illnesses or “colds” each year), (2) overlap between COVID-19 symptoms with other common illnesses, (3) screening for and recognizing symptoms during drop-offs only conditionally possible, (4) lack of understanding by parents, (5) parents hide symptoms of illness, (6) lack of support from physicians (i.e., official certification is no longer issued), (7) handling symptomatic child and asymptomatic sibling, and (8) summer vacation of families in high-risk areas (as defined by the Robert Koch Institute).

##### Symptomatic Staff

ECPs were not permitted to work when showing symptoms. All child-care managers incorporated this measure without exception. When symptoms were present, their employers arranged a testing appointment and staff self-isolated at home until a negative test result was available. Child-care managers alluded that in pre-pandemic times employees worked with mild symptoms. However, due to COVID-19, child-care managers took all kind of symptoms seriously and clarified them thoroughly. Accordingly, ECPs stayed absent longer than usual as awaiting the test result took time or alternatively, when no test was undertaken all symptoms had to be subsided.

##### Routine Testing Strategy

In addition to occasion-related testing (i.e., suspicion of infection and/or symptoms), ECPs could be tested voluntarily and free of charge every two weeks outside working hours. Since this measure was newly introduced in August 2020 by policymakers, only some of the child-care managers reflected on this measure (*n* = 10). Only one of them responded positively and took advantage of this offer. Others criticized the following aspects: (1) implementation in free time (demand: during working hours), (2) independent and complicated contacting of test centers/physicians for appointments, and (3) long waiting times (making an appointment and appointment itself). Two child-care managers arranged on-site testing through the commitment of general physicians (including a mother of a child), which was much appreciated by all those involved.

#### 3.1.6. Restrictions in Play for Children

Preventive actions taken also related to the child-care program. Guidance documents for play included, on the one hand, regulations for personal toys and, on the other hand, the restriction of usage of play structures and/or equipment in the child-care centers.

##### Stricter Regulations for Personal Toys

According to child-care policies, bringing own toys was prohibited. Some child-care managers were persistent in the implementation of this measure, especially if this arrangement existed prior to COVID-19, thus they faced no challenges. Other interviewees recognized the necessity of adapting this measure to individual needs: (1) personal stuffed animals/toys remained permanently in the center or child-care manager’s office, (2) familiar items from home are allowed for settling-in children, (3) personal toys for insecure children attending child care for the first time after lockdown are permitted, (4) allowance of toys that can be sanitized or laundered (e.g., exclusion of wooden toys), (6) stuffed animals for “sleep children” are allowed and provided during sleep time, and (7) only infants and toddlers (i.e., <3 years old) bring personal toys as only older children understand the usefulness of this measure. One child-care manager took advantage of the pandemic-related regulations to launch a toy-free time project.

##### Reduction in Play Opportunities

Interviewees described measures taken to prevent children from playing together in confined spaces as extensive. Child-care managers eliminated play opportunities through various measures: (1) closing places where children come close to each other (e.g., cuddle corners, ball pits), (2) distributing toys between group settings, therefore fewer toys remain in one group setting, (3) reducing to only those toys that can be sanitized, (4) limiting playing areas due to fixed group settings (e.g., not using corridor area, gym only every other day, in outdoor area only certain areas), (5) reducing the number of children with whom to play. As a consequence of disinfection measures, some toys broke, therefore fewer were available. If the size of the outdoor area and the stock of play material for all group settings were sufficient, child-care managers considered these measures to be manageable.

#### 3.1.7. Redesign of Catering and Food Service

With the reopening of child-care for all children and the associated high number of children in contrast to emergency period, the redesign of the catering involved major changes under enhanced hygiene practices.

##### Preparation of Meals

ECPs and children no longer assisted in the food preparation. Kitchen staff and/or cooks portioned and pre-plated the meals for the respective group settings. In few centers, the food was delivered and an assigned staff member pre-plated the meals in the kitchen. A large number of child-care managers accentuated the absence of buffet form due to COVID-19, which changed the breakfast situation in particular as it was individually brought from home or prepared in the kitchen on children’s requests. Child-care managers admitted the changes in the preparation of the food, but highlighted the feasibility owing to the support of kitchen staff.

##### Serving and Distribution of Meals and Beverages

Child-care managers attributed the greatest challenge to the serving of meals. In normal operation, children partake in handling table and place settings as well as in the distribution of meals. Instead, ECPs served children pre-plated meals wearing PPE (i.e., face masks and gloves). Child-care managers frequently compared this meal service with the service in a restaurant, which strongly contradicted their sense of children’s participation and autonomy. Self-service by children could not always be avoided due to limited personnel resources available. Additionally, child-care managers introduced other measures targeting the temporal and spatial equalization: (1) increasing the space between children at a table, (2) using different rooms for different group settings (e.g., group rooms, corridor, and staff room), (3) staggering mealtimes to reduce occupancy, and (4) personalized seating. Not all centers were able to implement the measures due to insufficient space, furnishing, and staff. Beverage bars or stations were dispensed with. Children brought sealed bottles from home, which were filled by ECPs at the child’s request. ECPs set up regular drinking times and paid higher attention to ensuring that children drink enough.

#### 3.1.8. Extension of Hygiene and Cleaning Practices

Albeit maintaining a sanitary environment displays a standard in a child-care setting, child-care managers still undertook adoptions and extensions due to SARS-CoV-2.

##### Compliance with a Hygiene and Cleaning Conceptual Framework

All child-care centers are required to adhere to a conceptual framework for hygiene and cleaning, which specifies detailed in-house procedures for maintaining infectious hygiene. Prior to reopening, ECPs updated the plan appropriately and worked out all hygiene-relevant measures under facility-specific and SARS-CoV-2-specific aspects. The definite implementation was mostly guaranteed by a comprehensive cleaning plan. Child-care managers supplemented this procedure individually by the following measures: (1) appointment of a hygiene officer in the team as a supervisor and control authority, (2) assignment of a fixed staff member per group setting per day who carries out the hygiene measures, (3) allocation of a fixed time in the afternoon (approximately 45 min) for large-scale disinfection of the entire premises when children left, and (4) scheduling of a pre- and post-processing time as time buffers. Child-care managers reflected on hygiene as a team effort well-supported by the whole team. Since some ECPs are afraid of contracting COVID-19, they respect and follow these measures largely.

##### Adaptions in Hygiene Practices

Topics around hygiene measures focused on four areas: (1) ventilation, (2) laundry, (3) surface disinfection, and (4) respiratory hygiene/cough etiquette and personal hygiene.

(1) Since all interviews were conducted in summer, windows and doors were opened continuously as a matter of course. A child-care manager mentioned uncertainties in the use of fans indoors, as fans are regularly used when the air temperature is high, but due to COVID-19 fans were only used in rooms with no occupants.

(2) ECPs paid precise attention to which towels, blankets, pillows, or other materials children used during the day and laundered these items more frequently (daily to at least twice a week) as opposed to prior to the pandemic.

(3) Child-care center managers highlighted the increased frequency and intensity of high-touch surface cleaning and disinfecting. The areas predominantly described, focused on (a) eating/food areas, (b) sanitary facilities, (c) toys, and (d) other contact surfaces or items (e.g., doorknobs, light switches, and stairways). Preceding and/or following lunch, staff cleaned and disinfected all tables and chairs, especially when they consecutively served different group settings in the same place. Cups and glasses were rinsed several times a day. ECPs cleaned and disinfected all contact surfaces in the bathroom after used by a child. Similarly, special attention was placed on sanitizing toys between usage, especially when exchanged between group settings. A child-care center manager described that due to a multi-story design, the stairwell needed to disinfected several times a day. In addition to these occasion-related and targeted disinfections, ECPs conducted routine, large-scale disinfection measures of all high-touch surfaces in the morning/at lunchtime/or in the afternoon.

(4) Child-care managers reported that many children already internalized these reinforced behaviors from home during the lockdown period. Only infants and children with special needs showed some difficulties and required assistance. Accompanying all children to the bathrooms and assisting in washing hands after drop-off could be easily introduced. However, monitoring all children during the daily routine was not always feasible due to lack of staff. Increased and more thorough hand hygiene by children over the day was supported in individual facilities by minor structural adjustments (e.g., adjusting the height of towel holders).

##### Modifications for External Cleaning Services

The majority of child-care managers particularly expressed a grown awareness of how accurate the cleaning services cleaned, the need to check whether certain cleaning requirements are met and if necessary, to rework them.

#### 3.1.9. Focusing on Occupational Health and Safety (OHS)

Child-care managers carried out protective behaviors and used PPE. In order to adopt a holistic OHS approach, they were additionally supported in assessing human and organizational aspects before reopening.

##### Protective Behavior and Personal Protective Equipment (PPE)

All adults wore masks during drop-offs and pick-ups and when a distance of 1.5 m could not be maintained. Interviewees described keeping physical distance between parents and staff as unproblematic. Yet, keeping physical distance between staff could not consistently be practiced in these situations: (1) breaks (i.e., break room too small), (2) carpooling, and (3) communication. Child-care managers who were generally not assigned to a fixed group setting wore a mask in the entire center, staff designed to a fixed group setting only wore a mask in common areas (e.g., corridor). Partially, ECPs paid attention to whether high-risk groups were present in the center and how each staff member felt comfortable and safe. Participants clarified that maintaining distance and using PPE during the interaction with children felt utopian and quixotic since child work necessitates physical proximity.

##### Individual Risk Assessment/Occupational-Medical Health Examinations

An occupational health examination was conducted in the majority of centers to evaluate the individual risk of employees for severe COVID-19 progression. Child-care managers were advised by an occupational physician (or general practitioner), who performed the classification into high-risk groups. Some child-care managers carried out the classification on their own based on provided templates by the health department. After individual consultation with physicians, high-risk groups worked from home (e.g., reworking concepts) or returned to work requiring additional protective procedures (e.g., no interaction with children in the center, consistent wearing of masks for the entire staff). A minority mentioned that no individual risk assessment took place (e.g., very young team).

##### Workplace Risk Assessment

Child-care managers received a template from the German Statutory Accident Insurance in order to prepare a workplace risk assessment. They identified working situations causing high potential of SARS-CoV-2 transmission and clarified which additional measures can or should be implemented. In the majority of centers, the entire team carried out this workplace risk assessment adapted it to individual conditions (e.g., centers with many children with additional support needs). Only in a few centers, professional occupational safety experts facilitated on-site support as a consultant.

### 3.2. Research Question 2: How Do Child-Care Managers Perceive and Evaluate Preventive Measures and the Changes Associated?

In addition to the descriptive accounts of introduced measures, child-care managers also spontaneously started to express a normative perspective on measures.

#### 3.2.1. Vagueness and Ambiguity of Measures

The interviews revealed that individual recommended measures were interpreted differently by child-care managers and, conversely, implemented differently. For example, the recommendation to have only one adult accompany the child to the child-care center caused confusion and was interpreted ambiguously: one adult in the current drop-off situation vs. one adult continuously over the period of restricted operation. Scope of interpretation remained also for the measure’s underlying rationale: e.g., why is the duration of child care reduced? Child-care managers initially reported ambiguity to this question, until most legitimized the measure with having additional time for cleaning measures. For these reasons, interviewees considered the measures as too vague and indeterminate with too many degrees of freedom. Within child-care policies, recommendations were made, which in the end were undermined by the reference to negligibility due to staff shortages, which in turn is a well-known problem in the child-care setting. ECPs lost track of currently applicable measures and were confused due to the flood of information, as well as rapid short-term changes. A clear guideline without complex medical terminology from one consistent source was missing.

#### 3.2.2. Reasonableness of Recommended Measures

Some child-care managers were highly critical of individual measures and/or their resulting consequences. By far the most negatively appraised was the change in the educational concept of the centers (see category difficult compatibility of early childhood pedagogy and infection prevention). Other criticisms mostly referred to the questionability of whether the measures actually reduce the risk of exposure or whether the measures are suitable in practice. The measures seemed to be too theoretically based. Interviewees specifically identified lacks in reasonability in the course of the following topics: (1) reduction in the duration of child care per child; (2) group settings, absence of sibling regulations and personnel flexibility; (3) symptomatic children, no need for a medical certificate; (4) strict personal toy regulations; (5) lunch/food service, physical distance and personalized space; (6) contamination despite strong cleaning and hygiene measures; and (7) measures for children with special needs. (1) The reduction in child-care hours did not lead to fewer children attending. Therefore, this measure failed to achieve the goal of reducing the risk of infection by having fewer children at one time and smaller group sizes, since most children were only picked up a little earlier than usual. Child-care managers felt that it made no difference whether they cared for the children 10 h more or less. However, some child-care managers still considered the reduction in child-care hours to be favorable, as more time allocated to hygiene and cleaning measures was available. (2) At the time of reopening, there were no clear guidelines on how to handle siblings (respectively children from the same household) in group settings. Siblings could be grouped in different group settings within one center (e.g., for educational reasons), which ultimately created a connection across group settings and made the division into group settings obsolete. In general, group settings reduced flexibility and the exchange of staff. Many of child-care managers experienced a challenge by implementing this measure. Nonetheless, child-care managers were aware of the advantage of group settings: In the event of a COVID-19 infection only the respective group setting was ordered to close. (3) Child-care managers classified the measure of allowing parents to attest children’s symptom freedom as unfavorable, and aimless. Interviewees argued how to ensure the health and safety of everyone in the center if no official certification from physicians is required. (4) The prohibition of private toys put a big question mark over the benefit in terms of IPC, as bringing other personal items such as drinking bottles and backpacks were permissible. (5) A high number of child-care managers considered the measure of creating distance between children during meals to be unreasonable, since children are allowed to play with each other throughout the day. Due to the lack of sufficient furniture, personalized seating during meals was unrealistic. (6) Interviewees emphasized that contamination of body fluids cannot be avoided in spite of highly differentiated hygiene measures, it must expected in child care. (7) Participants considered the measures for children with special needs or disability to be too undifferentiated and implied that the development of COVID-19-related measures neglected these children.

#### 3.2.3. Discrepancy between Measures in Public and in Child Care

Interviewees unanimously perceived a discrepancy between general public health and social measures and the OHS measures in child care; thus, considered the latter to be stricter. For example, participants described less restrictions in accessing public places (e.g., playgrounds, sports activities, and swimming pools) in comparison with the consistent and precautionary maintenance of preventive measures in the child-care sector. Interviewees could not comprehend this decision at the political level. This inconsistency led to families being able to meet in private which intermingled group settings in the center. Accordingly, interviewees felt that the strict preventive actions inside, and the less strict regulations outside the child-care center were conflicting. Thus, their great commitment in the child-care center was superfluous, as they could no longer guarantee traceability and ensure OHS due to external circumstances. If families did not have to adhere to these structures in a private context, parents and children lacked understanding for the stricter rules in child care. Interviewees criticized inadequate and unrealistic media coverage of the working and child-care conditions under COVID-19. Hence, the development and description of recommended measures were glorified and the actual, difficult, and stressful implementation by ECPs inadequately represented.

#### 3.2.4. Perceived Benefits due to the Measures

Beyond the negative evaluations, individual measures were evaluated positively and child-care managers considered to maintain several changes in normal operation. Through drop-offs and pick-ups outdoors, children became more independent in getting (un)dressed. Interviewees experienced these procedures as easier for ECPs and children; and perceived children to be more relaxed, as there was no parental pressure. The fixed schedule times led to better orientation for ECPs. Child-care managers observed that the group feeling strengthened within a fixed group setting and that permanent compositions were beneficial to provide ECPs and children security in an uncertain time. Introverted children and large child-care centers in particular profited from the group concept.

### 3.3. Research Question 3: How Do the Preventive Actions Taken Affect the Work Characteristics and Their Interpretation?

Navigating child care through COVID-19 and thereby balancing requirements of measures, the risk of SARS-CoV-2 exposure, as well as continuing child care resulted in numerous changes in work characteristics and therefore increased the work demands.

#### 3.3.1. Work Organization and Workload

##### Reopening Preparations: Increased Workload and Expenditure of Time

It became apparent that a considerable amount of organization needed to be accomplished even in the days and weeks before reopening. Child-care managers and their teams were urged to customize the pick-up and drop-off schedules and revise the hygiene and pedagogical concepts. Well-functioning open pedagogical concepts (i.e., children move around freely according to their own preferences) were converted into closed group concepts. A common guiding principle for drawing up plans was to strive for the most possible for children despite pandemic circumstances. ECPs adapted the physical space to meet the requirements of measures. This comprised the conversion or redesign of rooms as well as the redistribution and provision of play materials and furniture, which sometimes led to a financial burden. Interviewees described this preparatory work to be strenuous as they aimed to offer equal educational opportunities to each group setting. Some child-care managers also reported that it was time-consuming to obtain PPE (i.e., ongoing shortages), to acquire technical equipment and to learn how to use digital media as a means of communication.

##### Daily Working Routine: Increased Workload and Expenditure of Time

Child-care managers frequently discussed the major changes in the overall daily routine, which exceeded the workload compared with normal operation. ECPs dedicated compelling energy and resources with the aim to assure children a good day care routine in addition to implementing the measures. These tasks were particularly highlighted: (1) drop-offs and pick-ups, (2) hygiene and cleaning, (3) information and communication, (4) meal situation, and (5) leadership tasks. (1) The drop-offs and pick-ups resulted in additional work for both ECPs: for the person who picked up the children, as well as for the person who remained in the group with other children during this time. Tasks such as washing hands upon arrival and getting (un)dressed, which were otherwise the responsibility of parents, resulted in considerable additional workload. Some child-care managers of large centers reported longer distances to the entrance door, which intensified the effort again. (2) The increased hygiene requirements were omnipresent, described as the greatest burden and time-drain throughout the entire day, in contrast to the drop-offs and pick-ups, which had a determined timeframe. Interviewees expressed stress to constantly accompany children to the bathroom. (3) Keeping all parents and the team on a common level of knowledge, reminding them of measures and explaining new developments (e.g., legal regulations) extended the workload (see also categories “changed interaction with parents/within the team”). Additionally, it took time to keep in touch with children who were not yet attending the center (e.g., due to parents’ fear of infection). (4) ECPs’ workload in the meal situation increased since children were no longer actively helping. (5) Leadership tasks intensified as child-care managers acted as a constant point of contact and source of information round-the-clock. Interviewees were confronted with growing demands from politics and society which forced them to make responsible decisions. This implied that child-care managers themselves understood and internalized the new regulations amidst information flood in order to be able to pass them on correctly, which was again work-intensive. The onboarding and training of new staff required more time than usual, as did administrative tasks. A child-care manager reported anxious employees who needed special attention and encouragement.

##### Noisy Work Environment

Since groups stayed in one room based on the regulation of the fixed group setting, there was an increase in noise pollution.

##### Overtime

Due to the short notice regarding the reopening of the child-care centers and recommended measures, interviewees reported a large number of overtime hours for the entire team, particularly including weekends.

##### Interruption of Work and Multitasking

Especially the modified drop-off and pick-up situation forced disruptions in the workflow and interruptions in the activities previously performed. Therefore, ECPs were regularly pulled out of their work both mentally and physically. The multitasking also expanded, as ECPs carried out additional tasks, particularly those relating to hygiene, simultaneously to the nursing activities.

##### Higher Physical Demands

During the preparation for reopening, child-care centers were converted with great physical effort. Child-care managers emphasized the lack of support from craftsmen or other external personnel, so that reconstruction was carried out by the team and was accompanied through physical strain.

#### 3.3.2. Work Content and Tasks Regarding Pedagogy

The themes on the impact on ECPs’ pedagogical work recurred throughout all interviews. Not only did the qualitative content of the work change, but also the quantity pertaining to how much pedagogical work was viable.

##### Settling-In of Children during COVID-19

The interviews took place during the common settling-in period around August: Young children start child care, others move from a younger group to an older one, and elementary school children leave child care. ECPs approached the original processes differently as they conducted settling-in interviews with parents outside by individual appointment and organized a separate room for settling-in visits. Child-care managers questioned teaching closed concepts with fixed group settings while wearing PPE and while knowing that this arrangement could be near-term modified. The change from one group to another was usually performed gradually (i.e., a two- to four-week period), but given the circumstances ECPs were compelled to abruptly change the group composition. Elementary school children left child care without a proper farewell to the disillusionment of the ECPs.

##### Intensified Hygiene Education for Children

Interviewees outlined a shift of focus in their educational work in which hygiene education is highly relevant. ECPs addressed COVID-19 and the pandemic as key issues in a variety of ways (e.g., in ‘morning meetings’, books, role plays, posters, songs, etc.). This included an increased emphasis on respiratory hygiene, cough etiquette, and hand washing practices. Some child-care managers perceived themselves as role models for children, balancing justified caution without fear of social contacts while still building interactions.

##### Enhanced Pedagogical Attentiveness and Sensitivity

Following reopening, child-care managers and their teams had to react adequately to children’s different experiences in the previous months, as well as to their reactions to the new measures (e.g., group settings) and the resulting needs. The children’s responses were individual which required diverse pedagogical reasoning and actions by the ECPs. Occasionally, anxious children were accompanied more closely, or insecure children had to be reacclimated to child care. Other children had the desire to share their experiences, so that ECPs had to respond intensively to their accounts. Participants described it as arduous to meet everyone’s needs in this regard, because other children abdicated and did not want to bring up the topic of the virus at all (anymore). Not only was increased pedagogical attention on site imperative, but keeping in touch with children who stayed at home (e.g., concerned parents) also brought a new scope of content to the forefront of child care.

##### Difficult Compatibility of Early Childhood Pedagogy and Infection Prevention

A common view was that measures influenced or even hindered the usual development-oriented and child-centered work, so that preventive measures often focused only on the infection prevention. Finding a proper balance between infection measures and pedagogy and their competing goals was difficult. Child-care managers stated that as soon as it is a matter of children’s physiological or social needs that require immediate satisfaction in order to ensure the emotional well-being (e.g., grief, food and thirst), the balance is tipped, and the measures take a back seat. Depending on the situation, infection prevention or pedagogy came to the fore, and a perfect symbiosis between the two hardly seemed possible. Overall, child-care managers described the new work as a direct way of working with many instructions and prohibitions for the children as well as for their team. Some interviewees experienced the group settings as problematic from a pedagogical point of view. The associated lack of the open concept restricted the cross-group work or joint celebrations (e.g., summer festival, farewell to the school children), which are perceived as important in child care. All child-care managers felt a great need to be able to offer children pedagogical work despite infection protection, but they also reached their limits. Thus, pedagogically versatile, and uninterrupted work was limited given that group settings mostly remained in one room and time-consuming additional measures (e.g., cleaning toys and escorting children to the washroom) were present. Therefore, not only were children confined in their possibilities (e.g., restriction of toys, friends in other group settings and shortened opening hours), but also ECPs were restrained in the extent to which they could pursue their usual pedagogical work with passion. An overriding goal for educators is to teach children independence and participation. Child-care managers reported, for example, that the meal situation does not correspond to the actual pedagogical understanding; it is strongly opposed to it and ECPs perceived the implementation as burdensome. Similarly, accompanying the children to the bathroom was experienced as a form of control that contradicts the actual concept of the children’s independence and autonomy.

#### 3.3.3. Social Interaction and Cooperation

Physical distancing and reducing the number of interactions and/or contacts affected the interaction with parents as well as within the team. Child-care managers narrated that new forms of communication and cooperation had to be developed, tested, and adapted. Moreover, not only were the forms of communication modified, but the content, frequency, and dynamic as well.

##### Changed Interaction with Parents

The extent to which the interaction with parents changed depended on the initial reaction of parents to the reopening. Child-care managers described a broad spectrum of positive to negative parental reactions: (1) joy and relief about the reopening, especially due to the double burden of child care and work, (2) acceptance of the new measures and cooperation despite organizational challenges, (3) fears and anxieties about sending children back to the child care despite a risk of infection (e.g., children with special needs, large families), (4) anger and irritation due to new regulations, especially the reduction in child-care hours and the resulting restructuring of working hours as well as stricter regulations on symptoms of illness, and (5) disappointment and lack of understanding of essential workers (i.e., full scope of child care during the emergency child care).

Depending on the respective parents’ reaction, the interaction turned out to be easier or more difficult. Child-care managers reflected that parent meetings were cancelled and/or the interaction with parents was limited to a minimum, thus only crucial themes were discussed. However, they also emphasized the effort not to lose touch with the parents of the children who stayed at home. The dominant form of digital communication was e-mail, followed by messaging services. Some child-care managers reported the following difficulties in e-mail usage: (1) prior paper correspondence, needed to change at short notice, (2) not all parents provide e-mail addresses (privacy concerns), or (3) parents do not have e-mail addresses. Other digital forms of communication included video conferencing, and in one case a mobile application. Other child-care managers preferred telephone conversations, or face-to-face interaction with physical distance and face masks in outdoor spaces, especially for topics such as settling-in. Parent councils, which coordinated communication, had a supporting function in many facilities and thus relieved child-care managers. Nevertheless, interviewees addressed a variety of problems: (1) keeping all parents on the same level of information was more challenging than usual (i.e., child-care managers had to be more (pro)active), (2) parents experienced fewer opportunities for participation, (3) parents had to be informed more spontaneously and at shorter notice about upcoming changes, which in some cases required child-care managers’ constant availability, (4) preparing complex written information for parents of non-German-speaking origin was difficult, since these parents often depend on personal conversations with interpreters, and (5) a lack of contact information, e.g., from families from refugee housing units, impeded to pass on information. Topics that predominantly raised discussions with parents included (1) symptoms of illness, (2) brevity of information, (3) reduction in child-care hours, and (4) toy regulations.

##### Changed Interaction within the Team

Child-care managers cancelled regular team meetings across group settings and substituted these by holding individual or smaller team meetings. Digital communication forms were introduced (e.g., video conferences, messaging services). However, challenges emerged, as some child-care managers and ECPs were less familiar with digital communication and/or only had access to private devices. Face-to-face interactions took place at a distance outside or in larger premises inside (e.g., gymnasium). A common view among interviewees was that this limited and non-personal type of communication was more complex, and did not comply with their expectations of interaction in the workplace. Further, child-care managers emphasized a change in the content of communication as organizational arrangements took over time for pedagogical topics.

### 3.4. Research Questions 4: How Do the Preventive Measures Influence the Well-Being of Child-Care Managers and Their Staff?

#### 3.4.1. Mental Underload due to the Absence of Children

One interviewee outlined that employees felt unchallenged due to only few children attending child care in their center. It was described as an unfamiliar and unsatisfactory condition, which they never experienced until now.

#### 3.4.2. Worry about Children Staying at Home

Some children remained at home even after reopening child-care centers. According to a few child-care managers, ECPs were worried about certain children, especially if they had not yet been supervised by the youth welfare office. Thoughts revolved around children’s well-being and the experienced lack of control caused psychological discomfort.

#### 3.4.3. Emotional Exhaustion, Psychological, and Physical Strain due to Extra Work and New Responsibilities

Child-care managers signified emotional exhaustion by describing feelings of being emotionally overextended and drained by work; and not having sufficient or adequate resources at disposal (e.g., time, staff and flexibility) to meet current work demands. Reasons for experiencing strain were numerously stated: (1) short-term nature of measures, (2) information overload, (3) change in the pedagogical concept/work and the demand to represent a concept that does not correspond to personal ideas, (4) high workload including new tasks and multitasking and therefore a huge investment of resources (e.g., extra effort and energy), (5) perceived increased responsibility for the health of children, employees and parents, and (6) being faced with important decisions including conflicting opinions. Child-care managers’ emotional exhaustions manifested by physical fatigue as well as a sense of feeling irritable and psychologically exhausted.

#### 3.4.4. Anger due to the Lack of Reward and Appreciation

The topic of social appreciation emerged during most interviews, which tended to be expressed cynically. In this context, child-care managers compared their own occupational group with others (e.g., teachers, nurses and retail trade) and concluded that others received greater thanks and appreciation in media and public. Additionally, child-care managers denounced that child care is not acknowledged as an educational institution and is perceived wrongly in public. Hence, they wished for greater respect and understanding for their work. This assessment led to negative affectivity such as anger and frustration, as they recognized excessive workload combined with a high risk of infection in their work environment (i.e., reduced preventive behavior with younger children) that was not commensurate with the reward.

#### 3.4.5. Reduced Psychological Sense of Community and Identification at Work

During the interviews, it became evident that child-care managers perceive their child-care center as a large community of educators, children, and parents with a sense of identification. Interviewees argued that due to the pandemic-related measures, relationships between all three parties suffered. They experienced the parental exchange as disruptive to the relationship owing to frequent discussions. Fixed group settings forced ECPs not only to lose contact with other children and colleagues, but also hampered the feeling of belonging and connection to a larger social collective. Child-care managers described the loss of humaneness, sense of well-being, and shared experiences, and ultimately rated this as gloomy. Moreover, a few child-care managers implied that they will continue to suffer from that ramification in the future until all relationships are restored as before.

#### 3.4.6. Feeling of Being Left Alone and Loss of Control

Child-care managers felt forgotten by the legislators with the recommended measures. They had no opportunity to participate in the development of the measures initiated at a higher level and no support in the implementation of the measures. Consequently, it left them with a feeling of having no control. They also lacked the backing for their own decisions with the high level of responsibility in the child-care center. Interviewees felt left in the lurch and perceived this as incriminatory.

#### 3.4.7. Perceived Risk and Fear of Infection

Child-care managers and their team were aware of the fact that a risk of infection in the child-care sector cannot be avoided and considered this as an occupational risk that must be accepted. It was experienced as stressful to perceive the risk of infection, to want to protect oneself/others and concurrently to support all children. In this light, interviewees remarked not agreeing to reopen educational institutions despite cases of corona infections and the existing risk for ECPs in order to relieve parents. In addition to perceived risk, child-care managers considered the following as amplifiers of risk-related insecurity and uncertainty: (1) transmission events among children unclear, (2) opaque management of illness symptoms, (3) upcoming “wave” of disease in fall/winter, (4) inadequate PPE, (5) families on vacation (in risk areas), and (6) children in contact with an infected person. Participants expressed a strong fear of getting sick, speaking for themselves and their team, especially if classified as high risk. Since child-care managers were the only individuals in the center who had connections to all group settings, they were anxious about transmitting SARS-CoV-2 to all groups. The constant fear of infecting relatives and family was an omnipresent theme.

#### 3.4.8. Fears and Insecurity about the Future

The lack of knowledge and of planning as to what will happen in the coming weeks or months gave child-care managers a feeling of insecurity. The greatest fears for the future lay in these looming anticipated threats: (1) withdrawing proven infection control measures (e.g., group settings) and returning to a fully open operation, especially considering upcoming “winter illnesses”, (2) reoccurring high workload if measures change, (3) being (un)able to implement new measures quickly and well, (4) guaranteeing a satisfactorily settling-in time for new children, and (5) staff absenteeism due to high workload and/or illness and no guarantee of child care.

## 4. Discussion

### 4.1. Summary of Main Findings

The present study explored the implementation of COVID-19 preventive measures in child care, their appraisal, its impact on work characteristics and ECPs’ well-being from a qualitative perspective. A total of 27 child-care managers participated in interviews. Firstly, despite a high degree of heterogeneity in the child-care centers, the majority of recommended measures were feasible through the work engagement of ECPs, yet implied noteworthy effort. Secondly, the appraisal of measures revealed that critical voices dominated. This not only may hamper the successful implementation and compliance, but also can moderate the linkages between work demands and strains. Thirdly, the results indicate that demanding work characteristics that were pre-pandemic inherent to the child-care industry (i.e., work organization and workload) were exacerbated due to COVID-19 and the aforementioned preventive measures. We identified new psychosocial work demands pertaining to the change in work content and commensurate limitations in social interactions that negatively affected the work environment. Fourthly, this study revealed broad aversive psychological outcomes for child-care managers and their teams, which may put them on a trajectory for longer-term health issues considering the vague present and unforeseeable COVID-19 future. Given the qualitative and cross-sectional study design, we cannot establish associations between constructs.

#### 4.1.1. Practical Implementation of Preventive Measures and Associated Changes

A paucity of research studies examining the implementation of preventive measures in social care during COVID-19 exists. Previously only two studies addressed the feasibility and ease of implementation in educational settings during the reopening phase, yet these were across primary schools [74] as well as across primary and secondary schools [75]. A previous study with German child-care managers conducted during the emergency period supports the notion of extra effort due to preventive measures [76]. In our study, major changes after reopening referred to the measure of fixed staff compositions (i.e., group settings) that imposed staff shortages and inflexibility, which in turn increased the mental and physical effort for individual professionals. The stricter handling of symptomatic management further complicated adequate staffing, but also required additional mental effort when discussions with parents were required. The greatest physical effort lies in the reinforcement of intensified hygiene and cleaning practices present throughout the day. Therefore, the preventive measures to combat COVID-19 not only entailed short-term implementation efforts but had medium-term and long-term consequences. They also changed the physical, social, and organizational work characteristics of the child-care profession contributing to increased work demands. Moreover, a recent study shows associations between infection control programs in the workplace and mental health. Thus, workers that could not work remotely (e.g., ECPs) and whose infection control needs were not met, had the highest prevalence of anxiety and depressive symptoms among all workers [77].

Although it was feasible to implement preventive measures, it is vital to consider two issues: first, the feasibility does not necessarily imply a comfort or easiness in implementation, nor that it was readily implementable for professionals. Much more, that it was made feasible by ECPs’ efforts and the investment of available resources. In this vein, interviewees commonly displayed high occupational commitment during the reopening phase which accords to international literature with Chinese kindergarten teachers during COVID-19 [78]. From an infection prevention led perspective, it matters which measures are accomplished and how, but from an occupational health psychology perspective, what the consequences are for staff, too. Second, rapid and successful implementation depended on child-care centers’ ability and availability of resources (e.g., financial, information and personnel). Hence, a multitude of facilitators (e.g., adequate staffing, teamwork, personal strength and parental cooperation) and barriers (e.g., low engagement of parents, insufficient space, and materials) exist.

#### 4.1.2. Perception and Evaluation of Preventive Measures

According to the cognitive appraisal theory, the perception and evaluation of preventive measures served as a primary appraisal. ECPs determined whether the political expectations and the situational demands were harmful, threatening or challenging and therefore potential stressors [79]. We found that professionals perceived the preventive measures to be stressful events, which involved effort. Research suggests that this cognitive appraisal displays an underlying mechanisms that is essential for individual adaption to work settings [80]. Hence, a difficulty in adapting to COVID-19 circumstances reflected itself in interviewees’ negative accounts regarding work characteristics and work demands.

Implementation theories state the necessity of attributing meaning and committing oneself into a collective action to contribute to an effective implementation [81]. Albeit our sample displayed high occupational commitment, it is unclear whether child-care managers fully comprehended the array of preventive measures. Interviewees doubted the reasonableness and noticed a discrepancy to measures in public as they were deemed less rigorous. The comprehension was further complicated when measures were not properly understood due to diverse reasons (e.g., information overload, technical terms, different sources and dynamic changes) and were perceived as ambiguous. Similar to previous studies, this study therefore echoes the importance of cognitive and affective factors in adopting preventive measures during COVID-19. Studies show that the more knowledge, political trust, and the less negative an individual appraises the COVID-19 situation, the more likely the person will adopt preventive measures [82,83]. Hence, the critical perception of our participants may have hampered the implementation of preventive measures and hindered the accommodation of other changes within the workplace.

#### 4.1.3. Impact on Work Characteristics and Work Demands

We identified critical work characteristics (e.g., overtime, noise, interruption of work, multitasking and physical demands) that are consistent with previous quantitative findings (i.e., pre-COVID-19) on key psychosocial stressors [12,13,14,15,16,17,18,19]. This alignment between the existing literature and our exploratory findings suggests that high workload and poor work organization among this workforce are generalized to both scenarios, before and during COVID-19. Similarly, we replicated pre-pandemic findings of a qualitative study focusing on leadership-related work stressors in child care [22]. In our study, child-care managers were burdened by increased demands imposed by politics and society, the intensive volume of information as well as high responsibility. The COVID-19 crisis might exacerbate the difficult leadership role of child managers as crisis management requires specific skills (e.g., group decision-making, collective leadership) imposing new stressors [84,85]. It is conceivable that work characteristics regarding work organization deteriorated more or less strongly depending on implementation efforts, individual’s appraisals, as well as the resources present [85]. This coincides with other COVID-19 studies among other occupation groups in the health and social care sector, such as teachers [86], nurses [87], and outpatient caregivers [88].

Beyond work organization, a new salient theme emerged around work design: the change in work content and tasks as well as its decision latitude. Albeit the general task variety increased conjointly to workload due to the implementation of preventive measures (see also categories “reopening preparations” and “daily working routine”), other task characteristics are concerning. ECPs were not able to accomplish pedagogical work as the significant task in their profession as well as the nature of tasks linked. Thus, ECPs lacked task autonomy, which confers the absence of freedom to make work-related decisions [89]. The presence of COVID-19 including preventive measures decided and controlled how and when pedagogical work was carried out. Therefore, preventive measures reduced the degree of independence concerning the (1) content of task (e.g., less/other educational work), (2) work-scheduling (e.g., more rigid time management necessary), (3) work methods (e.g., no children participation and open pedagogical concepts) and (4) decision-making (e.g., needs of children vs. infection prevention) [90]. We moreover posit that the task significance was reduced, since nursing activities and the care for children could not take place to the usual extent and manner. Thus, other “illegitimate” tasks due to preventive measures created a threat to one’s professional identify [91] and may have reduced the meaningfulness in the work. As meta-analytic evidence and systematic reviews suggest lack of task autonomy and loss of control might further negatively affect job satisfaction, anxiety, stress and emotional exhaustion [92,93,94,95]. In adherence to the JDCS model, having control over one’s work can weaken the negative impact of work demands on strain [96]. However, ECPs were confronted with high demands and low control which is most likely to result in work-related stress-reactions. In addition, the third work characteristic “social support” in the JDCS model was an issue. The identified combination of high demands, low control, and low support mirrors previous findings with early childhood educators [24]. We found that fixed group-settings and general restriction in social contacts (e.g., physical distancing) changed and reduced the social interaction which reflected psychosocial stressors. Literature shows that high-quality social interaction at work is generally vital for psychological and physical health, especially in uncertain, demanding COVID-19 times [6,85]. Particularly in child care, teamwork and support are pivotal job resources for supporting well-being [38,97]. ECPs meetings have been shown to boost satisfaction and indirectly buffer emotional exhaustion [98]. Against this backdrop, the lack of social interaction and support may have heightened the level of perceived strain as social support was not able to moderate the negative impact of high demands and low control [36].

#### 4.1.4. Effects on Early Childhood Professionals’ Well-Being

Interviewees perceived the burden of challenging work conditions and limited social interaction in conjunction with the steady desire to provide high quality of pedagogical work despite measures. As predicted by the JDCS model our interviewees suffered from well-being impacts because of high demands, low control, and low social support received.

Our study reiterates findings of pre-pandemic studies such as emotional exhaustion as well as general psychological and physical strain reactions [26,27,99]. Similarly, our results are consistent with survey findings which suggest that increased demands in the child-care environment during COVID-19 elevated feelings of emotional drain [55]. In particular, parent-oriented tasks are associated with higher levels of emotional exhaustion [100]. This might be a relevant factor since greater coordination with parents was needed due to the implementation of measures, and many child-care managers tried to accommodate parents.

Similarly, a high prevalence of effort-reward imbalance among the child-care workforce was observed prior to COVID-19 [19]; and moreover shown to significantly increase the risk of burnout including emotional exhaustion [101]. During the reopening phase, ECPs encountered higher efforts and demands. Likewise, our interviewees recognized the insufficient reward from politics and society despite higher efforts. Both may have maximized the disparate effort-reward ratio and elicited the emotion of anger in child-care managers as an immediate response [102].

The feeling of being forgotten as a child-care manager was identified in another qualitative study prior the pandemic [22]. Our interviewees felt overwhelmed by politics and employers, since demands (i.e., preventive measures) were inflicted without prior notification nor participation. How child-care managers achieved the implementation, was their sole duty, which was further exacerbated due to absence of a management team consisting of several individuals and the absence of high social support. Thus, one person, and another deputy at best, bore full responsibility in an unprecedented context. We revealed that ECPs lacked a psychological sense of community at work. This also includes decreased coworker/parental support, a lower extent and quality of interactions, as well as a reduced sense of identification and belonging [103]. Since child care itself represents a community [103], it has the capability to provide individuals an enhanced quality of life, feelings of security, and coping abilities [104,105]. The absence of a community feeling may have had adverse psychological effects on ECPs.

Besides demand-related well-being impairments, other COVID-19-related well-being impacts were identified. Not surprisingly, given that respiratory infections are common in child care [106], our participants perceived a risk and consequently described a fear of SARS-CoV-2 infection which was also reported in other studies [21,107]. This further inflicts a dilemma of ensuring their own self-care vs. providing adequate care for others which causes moral distress and has been described in the health sector [108,109]. In general, research to date has not fully determined the role of children in COVID-19 transmission [110,111,112,113] resulting in uncertainties for all concerned parties. The most recent meta-analysis suggests that nearly half of COVID-19 cases in children younger than five years are asymptomatic underpinning the need for surveillance in child care [114].

Current reviews on COVID-19-associated mental health effects concluded that infection-related fears (i.e., contracting and transmitting) are a significant source of stress and anxiety in the workplace, particularly for frontline workers [115,116]. Yet, strong future-related fears as expressed by our interviewees were not stated. This result might be due to the fast-changing work environment resulting from ad-hoc childcare-policy responses. Our qualitative approach allowed us to uncover this fear of the future, interestingly some reviews intentionally excluded qualitative studies which may have led to this neglect [115].

One other study reported ECPs’ concern or unease relating to children’s well-being in their home environment [55]. Prior to COVID-19, ECPs experienced control in the event of an alleged or ascertained danger to a child’s best interest at home. This is owed to the fact that all children attended child care as a matter of course since Germany operates universal child-care programs. During COVID-19 it has become more commonplace for parents to leave children at home, thus ECPs no longer can see all children. Several researchers highlighted the surge of family violence [117,118] and general detrimental effects on children’s mental health in the midst of COVID-19 [119], which further substantiates ECPs’ experience of worry.

### 4.2. Strengths and Limitations

This study serves as an initial investigation concerning adaption and functioning of ECPs under COVID-19 circumstances. The qualitative research method uncovered pandemic-related challenges and its psychosocial working conditions among the child-care workforce. We obtained a wide possible range of opinions as we interviewed child-care managers with diverse work experience and sociodemographic characteristics as well as different child center designs. The comparatively high number of interviews [120] originates from two aspects: First, a heterogeneous sample of child-care managers generously shared a variety of complex accounts until data saturation was reached. Second, we strived for broad objectives within this exploratory study.

With qualitative research designs come general methodological limitations. Therefore, we cannot exclude a selection bias due to the voluntary participation, which presumes that interviewees were generally interested in the topic and in being interviewed. Additionally, we solely recruited within one city in one German federal state. That might restrict the scope of shared views and fail to capture potential other relevant topics in rural areas and other federal states. Subjectivity in our interpretations may have differed from interviewees’ intended perspective. To reduce subjectivity and ensure the accurateness of data, the procedure of member-checking could have improved the validity [121]. Yet, two researchers independently analyzed the data, with additional guidance of a third when discrepancies occurred. We further presented rich descriptions of our research process and results as well as provided verbatim quotes in the Appendix A to increase the credibility and trustworthiness of our findings. The study only represents one point in time during the pandemic. COVID-19 and the responses are dynamic; therefore, other workplace factors and health-related outcomes might have been omitted. However, we aimed to adequately capture the ECPs’ first experience and therefore conduct interviews within the two months of the reopening phase in child-care centers. Lastly, we only focused on child-care professionals, other occupational groups may have experienced similar or different challenges at the workplace as a result of preventive measures. Work in child care requires the responsibility for implementing additional or other preventive measures due to the special considerations for young children. Hence, the generalization to other professional populations is limited.

### 4.3. Implications for Research

Overall, future research studies are well-needed since stress research has largely overlooked ECPs for decades [122,123]. Child-care providers as frontline employees enabled the continued functioning of society and economy during COVID-19 [124]. This highlighted significance may accelerate the expansion of research, but also lift the child-care workforce higher on the political agenda [125]. Due to vast research opportunities, we can only feature several. First, our findings call for further empirical proof by quantitative and longitudinal studies with larger, more representative samples. Ideally, studies should investigate the impact of task-oriented work characteristics and control (e.g., task significance and task autonomy) on ECPs’ strain reactions, since the reduced practice of pedagogical work seemed to be a major source of impaired well-being. Our findings should be moreover expanded for different pandemic stages and post-COVID-19. Second, future studies should determine ECPs’ job and personal resources to overhaul the outdated data on this topic. Subsequently, workplace health promotion interventions (behavioral and structural) need to be developed, evaluated, and incorporated in child-care settings, since such effectively proven programs are lacking but indeed needed. However, even if interventions can be developed, many problems seem to be of a structural nature (lack of staff and insufficient equipment) that need to be addressed at a higher level.

### 4.4. Implications for Practice

This study provides educators and policymakers with ecologically valid findings from the frontline. Given that child-care policies in the reopening phase varied across 28 European countries [125], a long-term key policy priority should therefore be a uniform, scientifically-based, and updated guideline for effective emergency or pandemic preparedness in child care. Within this guidance, reasonable efforts need to actively anticipate the mental health and well-being of ECPs and contemplate psychosocial side effects of preventive measures in their workplace. Generally, workers’ voices and their priorities for particular OHS and IPC measures and workplace conditions during a pandemic are rarely considered. We encourage fostering a participatory approach when developing measures and guidelines at a policy as well as science-level [107]. Experts from the field, who vary particularly in their professional experience, but also in their socio-demographic characteristics, can co-develop and co-design preventive measures (e.g., Citizen Science approach). Moreover, we recommend elucidating the underlying, scientific rational for why a preventive measure is endorsed to be implemented. Participation and thorough explanations will help to leverage commitment and sense-making processes; therefore, will support the adaption to the changed work environment. During a crisis, uniform recommendations from one source of information are needed to avoid ambiguity. Yet, external specialists can be available as advisors during the implementation and ensure that all measures are sufficiently understood; or if measures require physical efforts, external support should be provided. Informing parents, respectively clients or other stakeholders at work, about the necessity and usefulness of preventive measures should not be the responsibility of the employee itself, but political responsibility. However, it is clear that due to the complex nature of child care, an issue for preventive measures in practice is inevitable [125]: finding the right balance between goals of infection prevention (public-health focused), the reopening of education and care services and children’s needs (education-focused), work-family reconciliation (work-family and social-inequality focused), and the health and safety of ECPs.

Demanding work characteristics within the child-care sector that have been known for decades should be finally addressed as this pandemic has once again clearly elevated it. Structural characteristics explain almost a third of variance in perceived work demands and outweigh other characteristics [38]; thus, ensuring adequate staffing (e.g., sufficient number of staff and lower child-to-staff rations), the deployment of floaters, as well as an adequate physical workspace (e.g., noise protection measures and space for breaks) will positively influence the experience at work. Therefore, measures that target structural characteristics are a definite need. Finally, the appreciation of ECPs on the part of parents, politics, and society warrants more attention, which may have a chance of improvement through the experience of COVID-19.

## 5. Conclusions

Qualitative interviews shed light on the return-to-work consequences for child-care managers and their staff in terms of work conditions and well-being in the midst of COVID-19, a yet neglected domain. It is worth noting that the reopening of child-care centers has come at a cost for ECPs. The preventive measures imposed by child-care policies entailed wide-ranging challenges in their daily working activity as well as regarding the preparation of reopening. Interviewees doubted the reasonableness of the preventive measures and experienced difficulties in the comprehension of certain measures. Thus, critical work organization and workload in child care were exacerbated due to implementation efforts and COVID-19. High demands and low control paired with low social support affected ECPs’ well-being by negative strain reactions. It is apparent that ECPs have made an exceptional contribution in supporting children and their families throughout the pandemic and will continue to. However, what is less clear is how this vulnerable occupational group faced with adverse work demands and impaired well-being is supported. Our study presents the actual deteriorated psychosocial work environment in child care. Therefore, actions must be taken which delve deeper into pandemic-(un)specific stressors and interventions to ensure a healthy and safe work environment, even post-COVID-19.

## Figures and Tables

**Figure 1 ijerph-19-01739-f001:**
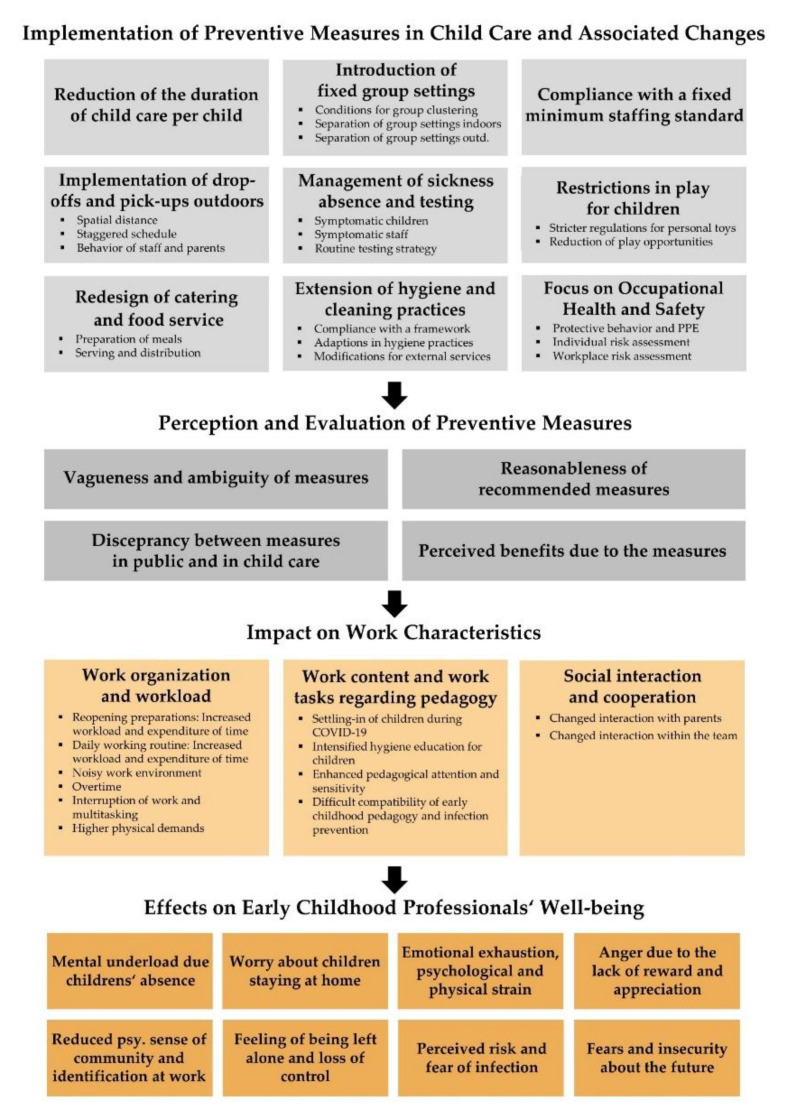
Graphic illustration of coding system with dimensions, main categories, and sub-categories as bullet points.

## Data Availability

The original data analyzed are available from the corresponding author on reasonable request.

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
