# Peer review of "The Impact of the Implementation of Preventive Measures Due to COVID-19 on Work Design and Early Childhood Professionals’ Well-Being—A Qualitative Study"

_ijerph, 2022, doi:10.3390/ijerph19031739_

Round 1

Reviewer 1 Report

Dear Authors,
the article is interesting. The literature review was very well done. The research process does not raise any objections. I appreciate qualitative research very much. The use of them is justified. The inference is also correct. I believe the article is fully suitable for publication.

Author Response

Thank you very much for your positive feedback and encouraging comment. We are glad about the support of our qualitative research design and your recommendation for publication.

Reviewer 2 Report

Comment on ijerph- 1489127

Title

  1. The title should reflect the major relationships in the manuscript. Implementation of preventative measures is a key construct, but it is neglected in the title. The title can be

 “The impact of Implementation of preventative measures of COVID-19 on work design and early childhood professionals’ well-being”.

  1. The complete manuscript is lengthy. You should do your best to make the manuscript more concise. I suggest writing the manuscript in an academic manner. For example, in your Abstract, you may articulate major relationships you derive from your qualitative studies.

Introduction

  1. Though the study is practice-oriented, you should still state the theoretical perspective and theoretical development up front.
  2. The research questions seem to be separate with each other.
  • You focus on childcare professionals. It is not clear how such focus implies for processionals in general and you also need to indicate the importance to adopt the preventative measures.
  • The discussion of work characteristics and work demands should be placed on prior research on these topics. For example, how prior scholars define work characteristics? What are the major antecedents of work characteristics? That’s to say, highlight the theoretical significance of your study through joining in a conversation among scholars, not just indicate what you intend to do.
  • Similarly, put forward the research question on the relationship between preventative measures and well-being in the academic context.
  • What are interlinks among these research questions?
  • Define the constructs in your manuscript.
  • Articulate why it is important to address these research questions.

Data analysis

  1. For your data analysis, please indicate how you address these questions.
  2. You should describe how the findings emerged from the data. Findings should be consistent with and reflective of data.

Results

  1. The results are detailed. Please give some attention to the validity concerns. For example, how do you ensure the content validity of your constructs?

Discussion

  1. The findings should contribute to theory development and provide insight into important professional issues. The authors should relate the findings back to the existing literature. I find that authors did quite poorly in the theoretical implication of the study.
  2. For practical implication, you seem to mix work up characteristics and work demand. Job complexity, information processing demand, problem-solving demand, skill variety and specialization are all work characteristics that represent work demand. You may need to specify the work characteristics so that readers can quote your work in their study. Since you do not make it clear, we do not know what can be taken away from your study in terms of theoretical contributions.

Reviewer 3 Report

It was a pleasure to review this manuscript on reopening child care programs during COVID-19.

Overall, I find this to be an interesting and well described study on a relevant topic. I recognize the authors’ hard work on this study, and I do not see any major shortcomings in the manuscript.

Author Response

Thank you for your very positive and benevolent evaluation. We sincerely appreciate the acknowledgement of the hard work we put into this study.

Reviewer 4 Report

I really enjoyed reading this article. It's a subject of great interest to me as a psychologist. The methodology is one I have used and I found it interesting to see how it was applied to this work.

Author Response

Thank you for your very positive and benevolent feedback and for emphasizing the academic interest in the topic of our study.

Round 2

Reviewer 2 Report

I still find many grammatical errors in the manuscript. For example,

  1. to be partly unreasonable, ambiguous and stricter than in public (see Abstract)
  2.  the emerge of (see Abstract)
  3.  child hood or childhood.

There are too many key words, but there is no the key word of preventative measure.

Please carefully proofread the manuscript before submission.  

Author Response

Thank you very much again for your feedback. We included "preventive measures" as a key word and deleted unnecessary other keywords. Furthermore, we carefully proofread the entire manuscript for English language and style. This was supported by a native English speaker who is now mentioned in the acknowledgements. As a result, we have corrected, for example, the identified grammatical errors in the abstract as mentioned in your comment.